# Comparing Intraperitoneal and Intravenous Personalized ErbB2CAR-T for the Treatment of Epithelial Ovarian Cancer

**DOI:** 10.3390/biomedicines10092216

**Published:** 2022-09-07

**Authors:** Naamit Deshet-Unger, Galit Horn, Moran Rawet-Slobodkin, Tova Waks, Ido Laskov, Nadav Michaan, Yael Raz, Vered Bar, Adi Zundelevich, Sara Aharon, Lubov Turovsky, Giuseppe Mallel, Seth Salpeter, Guy Neev, Kenneth Samuel Hollander, Ben-Zion Katz, Dan Grisaru, Anat Globerson Levin

**Affiliations:** 1Immunology and Advanced CAR-T Therapy, Tel-Aviv Sourasky Medical Center, Tel-Aviv 6423906, Israel; 2Department of Immunology, The Weizmann Institute, Rehovot 7610001, Israel; 3Department of Gynecologic Oncology, Tel-Aviv Sourasky Medical Center, Tel-Aviv 6423906, Israel; 4Sackler Faculty of Medicine, Tel-Aviv University, Tel-Aviv 6997801, Israel; 5cResponce^TM^ Company, Rehovot 7670102, Israel; 6The Hematology Laboratory, Tel-Aviv Sourasky Medical Center, Tel-Aviv 6997801, Israel; 7Dotan Center for Advanced Therapies, Tel-Aviv Sourasky Medical Center, Tel-Aviv University, Tel-Aviv 6423906, Israel

**Keywords:** CAR_T, ovarian cancer, immunotherapy, high-grade serous ovarian carcinoma, intraperitoneal treatment, ErbB2

## Abstract

High-grade serous ovarian carcinoma (HGSOC) is the most common type of epithelial ovarian cancer. The majority of cases are diagnosed at advanced stages, when intraperitoneal (IP) spread has already occurred. Despite significant surgical and chemotherapeutic advances in HGSOC treatment over the past decades, survival rates with HGSOC have only modestly improved. Chimeric antigen receptor (CAR)-T cells enable T cells to directly bind to tumor-associated antigens in a major histocompatibility complex-independent manner, thereby inducing tumor rejection. While CAR-T cell therapy shows great promise in hematological malignancies, its use in solid tumors is limited. Therefore, innovative approaches are needed to increase the specificity of CAR-modified T cells against solid tumors. The aim of this study was to assess the efficacy and safety of intraperitoneal (IP) versus intravenous (IV) CAR-T cell therapy in the treatment of HGSOC. We constructed a CAR that targets the ErbB2/HER2 protein (ErbB2CAR), which is overexpressed in HGSOC, and evaluated the functionality of ErbB2CAR on ovarian cancer cell lines (OVCAR8, SKOV3, and NAR). Our findings show that an IP injection of ErbB2CAR-T cells to tumor-bearing mice led to disease remission and increased survival compared to the IV route. Moreover, we found that IP-injected ErbB2CART cells circulate to a lesser extent, making them safer for non-tumor tissues than IV-injected cells. Further supporting our findings, we show that the effect of ErbB2CAR-T cells on primary HGSOC tumors is correlated with ErbB2 expression. Together, these data demonstrate the advantages of an IP administration of CAR-T cells over IV administration, offering not only a safer strategy but also the potential for counteracting the effect of ErbB2CAR in HGSOC. Significance: IP-injected ErbB2CAR-T cells led to disease remission and increased survival compared to the IV route. These findings demonstrate the advantages of IP administration, offering a safe treatment strategy with the potential for counteracting the effect of ErbB2CAR in HGSOC.

## 1. Introduction

Epithelial ovarian cancer (EOC) is the most lethal gynecologic malignancy [1]. High-grade serous ovarian carcinoma (HGSOC), the most common type of EOC, frequently presents at advanced stages (III or IV) with overt peritoneal metastases and ascites, resulting in high mortality rates. Most HGSOCs will respond to the initial surgical and chemotherapeutic treatment but the majority of which will reoccur, resulting in disease progression and mortality [2]. Novel therapeutic approaches are urgently needed to combat this deadly disease.

The ERbB family of receptor tyrosine kinases plays a key role in the tumorigenesis of many types of solid tumors, including—but not limited to—ovarian tumors. They promote tumor progression via cell proliferation, survival, migration, adhesion, and differentiation [3]. Overexpression and/or mutations in EGFR and ERbB2 are well documented in ovarian cancer and have therapeutic implications [4]. The rates of ERbB2 overexpression and/or amplification in ovarian cancer are variable [5], and while widely investigated, current attempts to target ERbB2 in ovarian cancer have been disappointing.

While cell-based immunotherapeutic modalities are common practice in hematological malignancies, with excellent clinical outcomes in various relapsing/refractory diseases [6], their use in solid tumors is limited. EOC is considered a “hot tumor” due to the presence of tumor-infiltrating lymphocytes that correlate with a good clinical prognosis [7], making it a potentially immunoreactive tumor type. However, immunotherapy has not been shown to provide a significant response in EOC [8,9]. Chimeric antigen receptor-T (CAR-T) cells are genetically modified T cells, isolated by the aphaeresis of peripheral blood for the treatment of various cancers. CAR-T cells are directed at tumor-associated antigens, which are self-proteins abnormally expressed by cancer cells [10,11,12]. The main targeted proteins for CAR-T cell-based treatments for HGSOC that are overexpressed in the tumor (e.g., hTERT, ERbB2, mesothelin, and MUC-1) [13] shows limited success due to the reduced number of targetable membrane antigens and their heterogeneous expression patterns, as well as difficulties in achieving effective concentrations of T cells at target tumor sites. In addition, potential antigen targets in EOC may also be expressed by normal tissues, suspecting them to be damaged by the circulating CAR-T cells.

The targeting of ErbB2 using CAR T-cells is supported by the prevalence of ErbB dysregulation in diverse solid tumors and the clinical impact of monoclonal antibody therapy directed against members of this family. However, the key obstacle to effective clinical translation is the risk of on-target toxicity owing to the lower-level expression of ErbB family members in many healthy tissues [14].

Since EOC is most often limited to the peritoneal cavity, it is reasonable to combine cytoreductive surgery with intraperitoneal (IP) chemotherapy for its treatment. Several studies reported a pharmacologic advantage for IP versus intravenous (IV) delivery of chemotherapy [15], with improved absorption and susceptibility of cancer cells, longer persistence in the peritoneal cavity, and refined chemotherapeutic doses that limit toxic side effects [16]. Therefore, we were intrigued to attempt a non-conventional IP delivery of CAR-T cells to overcome some of the limitations associated with their IV delivery.

In this study, we utilized CAR-T cells directed against ErbB2, which is highly expressed in EOC [17], and compared IP and IV administrations. In line with our hypothesis, the IP treatment not only improved survival, but also demonstrated an improved safety profile with fewer circulating lymphocytes, emphasizing its potential for the treatment of EOC. Moreover, ErbB2 expression levels in primary human ovarian tumors influenced the reactivity of ErbB2CAR.

## 2. Materials and Methods

### 2.1. Antibodies and Reagents

The antibodies for flow cytometry included anti-human ErbB2-allophycocyanin (APC) (Biolegend; 324408), anti-human CD3 (Biolegend; APC-300439 or pacific blue (PB-344824), anti-human CD8-PB (Biolegend; 301023), anti-human CD4-brilliant green (BG) (Biogems; 06111-40-100), anti-human CD45RA-APC (Biolegend; 304112), anti-human CCR7-Percp 5.5 (Biolegend; 353220), and anti-human cleaved Caspas3. For the preparation of activation plates, we used anti-human CD3 (Biolegend; 317326) and anti-human CD28 (Biolegend; 302934) (Invitrogen; PA5-114687). RetroNectin (Takara; T100B) was used for the preparation of pre-coated transduction plates. Transduced lymphocyte medium was supplemented with human IL-2 (Novartis-Pharma; 4764111 U57).

### 2.2. Cell Lines and Culture

The ovarian cancer (OC) cell lines included NAR, OVCAR8 (both kindly provided by Prof. Dan Peer from Tel-Aviv University), and SKOV3 (ATCC HTB-77). Peripheral human blood lymphocytes (PBL) were isolated from the blood of healthy human donors (obtained from the Israeli blood bank at Sheba Medical Center, Israel; Helsinki number 0803-17-TLV). These cells were cultured in RPMI 1640 supplemented with 10% fetal calf serum (FCS), 2 mM glutamine, and 1 mM sodium pyruvate (all from Biological Industries, Israel). The 293T (human embryonic cells; ATCC CRL-1573) and PG-13 (gibbon ape leukemia virus-pseudotyping packaging cell line; kindly provided by Ralph Wilson, Rotterdam Hospital) were cultured in DMEM supplemented with 10% FCS, 2 mM glutamine, and 1 mM sodium pyruvate. All media were supplemented with a mixed antibiotic solution containing penicillin (100 U/mL), streptomycin (100 μg/mL), and neomycin (10 μg/mL) (Biological Industries, Kibbutz Beit, Israel). Human primary cells purchased from Cell Biologics (Chicago, IL, USA) included cardiac microvascular endothelium H-6024, kidney epithelium H-6034, alveolar epithelium H-6053, and pancreatic epithelium H-6037. Red blood cells were isolated from the blood of healthy human donors (see “Single cell isolation” below). Each primary cell was thawed and propagated according to the supplier’s instructions. Cells were incubated in a humidified 37 °C incubator with 5% CO_2_, except for the PG13 cells, which were maintained in a 7.5% CO_2_ atmosphere. All cell lines were verified by PCR (HyLabs, Kibbutz Beit, Israel) to lack mycoplasma. The cells were frozen at low passage, and the number of passages after thawing was recorded. The cells were maintained in culture for no longer than 4 weeks, which corresponds to approximately 12 passages.

### 2.3. Construction of the CAR Retroviral Vector

The N29 monoclonal antibody against human ErbB2 was used as a source of the single-chain fragment variable (scFv) for construction of the ErbB2-directed CAR [18]. The ErbB2 scFv was ligated to a portion of the human CD28 costimulatory molecule (encompassing its cytosolic hinge, transmembrane, and inner signaling domains) followed by the activating domain of the human FcγR molecule (incorporating a tyrosine-based activation motif immunoreceptor) at the 3′ end. A GFP reporter gene was integrated by means of combined sequences of an internal ribosome entry site. The construct was sub-cloned into the pBullet retroviral backbone (Figure 1A).

### 2.4. Preparation of Packaging Cells

The 293T cells were transfected by Ca_2_PO_4_ with a GAG-POL pCL-Ampho retroviral envelope [19] and the pBullet-ErbB2-GFP. The retroviral supernatant was collected and used to stably transduce the amphotropic PG13 packaging cells. Infected cells were sorted by a BD FACS Aria III Cell Sorter to enrich CAR-expressing cells, after which sorted cells were expanded and frozen in aliquots.

### 2.5. T-Cell Transduction

Retroviral transduction of T cells was performed as previously described elsewhere [20].

### 2.6. Flow Cytometry Analysis

Transduction efficiency of ErbB2CAR was estimated based on green fluorescence protein (GFP) expression. The staining of cells with fluorescent antibodies was monitored by a BD CANTO flow cytometer. Data analysis was carried out with FCS Express software.

### 2.7. In Vitro CAR-T Cell Mediated Killing

The cytotoxicity of CAR-T cells was determined by a methylene blue staining-based assay. ErbB2CAR-T cells or untransduced (UNT) lymphocytes were incubated with target cells in a 96-well plate at effector:target (E:T) ratios of 1:1, 3:1, 10:1, and 30:1. The cytotoxicity of CAR-T cells was determined by an assay based on methylene blue staining, as recently described [20].

### 2.8. In Vitro IFNγ Secretion Assay

ErbB2CAR or UNT T cells were incubated for 24 h with target cells (ovarian cell lines or primary cells) at a 2:1 E:T cell ratio. Cell-free growth medium was collected and analyzed for interferon γ (IFNγ) secretion as previously described [20].

### 2.9. Luciferase Gene Transduction of NAR Cells

Lentiviral particles were produced in 293T cells by calcium phosphate-mediated transfection involving a three-plasmid expression system. The 293T cells were plated into 10 cm^2^ plates at 4–5 × 10^6^ cells/plate for 24 h. The transfection included a vector plasmid (pLB(N)CX-CMV-luciferase) at 10 μg, pUMVC (GagPol MLV) at 7.5 μg, and pMD.G-VSVG at 2.5 μg. The plasmids were mixed with calcium chloride (1.25 M) and HEPES-buffered saline and then added to the 293T cells for 6 h, after which they were replaced with fresh medium. The viral particles in the medium were collected 48 h later, filtered through a 0.45 μm membrane, and incubated with pre-seeded NAR cells in the presence of polybrene (8 μg/mL; Sigma-Aldrich, St. Louis, MO, USA). The viral supernatant was replaced by fresh medium after 4 h of incubation. Expressing clones (referred as to NAR-LUC) were enriched with 6 μg/mL Blasticidin antibiotic selection. Luciferase expression was analyzed by an IVIS Lumina series III imaging system (PerkinElmer, Waltham, MA, USA).

### 2.10. In Vivo Experiments

All animal experiments were approved by the Tel-Aviv Sourasky Medical Center (TASMC) ethics committee (see Ethics, below). The experimental mice were 10–12 weeks old NOD/SCID/IL2Rγ^−/−^ obtained from The Jackson Laboratories (Bar Harbor, ME, USA). They were maintained in a specific pathogen-free facility of the TASMC. A total of 2 × 10^6^ NAR-LUC tumor cells were injected intraperitoneally. After 18–23 days, ErbB2CAR or UNT T cells were administered to tumor-bearing mice by either IP or IV injection. The anti-tumor response was evaluated by following survival up to 92 days and by in vivo imaging detection of luciferase signals expressed by NAR LUC cells. Bio-luminescence imaging was performed following IP injection of luciferin (375 μg/kg) with the animal anaesthetized with ketamine (100 mg/kg) and xylazine (20 mg/kg) in accordance with institutional guidelines. The mice were monitored by an IVIS Lumina series III imaging system (Perkin Elmer, Waltham, MA, USA). For flow cytometric analysis, the mice were euthanized either after a 10% weight loss or 45 days from the initiation of the experiment.

### 2.11. RT-PCR

Total RNA was extracted from ovarian and splenic tissues. Single-cell suspensions were dissolved in Trizol Reagent (Merck, 82024 Taufkirchen, Germany) according to the manufacturer’s instructions. cDNA was produced with a high-capacity cDNA reverse transcription (RT) kit, and RT-PCR was performed with fast SYBR green master mix (both from Thermo Fisher Scientific, St. Louis, MO, USA) on a StepOnePlus real-time PCR machine (Applied Biosystems, Foster City, CA, USA). Changes in relative gene expression were calculated by means of the 2^−ΔΔct^ method, normalizing to the housekeeping gene, GAPDH. The primer sequences (IDT, Jerusalem, Israel) used for amplification are listed in Table 1.

### 2.12. Single-Cell Isolation from Mouse Tissues and Human Ovarian Tissues

Spleen and ovarian tissues from NAR-LUC-bearing mice were dissociated by crushing with a plunger. The cells were filtered over a 70 μm cell strainer and centrifuged at 1200 rpm for 5 min at 4 °C. Red blood cells were lysed by NH4Cl and NaHCO_3_ lysis buffer. Cell suspensions were washed with PBS, and the cells were analyzed with either RT-PCR or flow cytometry. An amount of 100 µL of peripheral blood samples from NAR-LUC-bearing mice were lysed with ACT lysis buffer, washed, and stained for flow cytometry.

Fresh HGSOC tissues were obtained from patients undergoing cytoreductive surgery. A 1 cm^3^ section of tumor was cut and sent to the cResponce^TM^ Company. The cResponce^TM^ platform enables prediction of tumor response to various drugs or drug combinations utilizing fresh tumor samples processed into cell cultures. The remainder of the tumor was dissociated into 1 mm^3^ pieces by means of a scalpel and subjected to enzymatic digestion solution containing RPMI supplemented with 1 mg/mL collagenase/dispase (Roche Diagnostics; 8305 Mannheim, Germany) and 20 µg/mL DNase I Catalog number: 07469 (Stemcell technologies, Seattle, WA, USA) for 30 min at 37 °C in a shaker. Subsequently, the digestion medium containing the remaining tumor pieces was filtered over a 70 μm cell strainer. The cells were centrifuged at 300 g for 5 min at 4 °C. Red blood cells were lysed with ACT lysis buffer, and single cells were subjected to a flow cytometric analysis.

### 2.13. Ex-Vivo Organoid Culture (EVOC) System

HGSOC tissues were sliced to 250 μm thick slices by means of a vibratome (VF300, Precisionary Instruments, Boston, MA, USA), placed in 24-well plates on metal grids with 0.5 mL of DMEM/F12 medium (supplemented with 5% FCS, penicillin 100 IU/mL with streptomycin 100 μg/mL, amphotericin B 2.5 μg/mL, gentamicin sulfate 50 mg/mL, and L-glutamine 100 μL/mL). Tissues were cultured at 37 °C, 5% CO_2_, and 80% O_2_ on an orbital shaker TOU-120N (MRC, Holon, Israel) at 70 rpm. On the following day, the tissues were treated with ErbB2CAR or UNT T cells for 72 h. The conditioned medium was collected for analysis of IFNy secretion. Upon completion, tissues were fixed overnight with 4% PFA, followed by formalin-fixed paraffin embedding (FFPE).

### 2.14. Scoring of Ex Vivo Organ Culture

Hematoxylin and eosin (H and E) staining was performed with an automated stainer. Immunohistochemistry staining via an automated stainer (BOND RX, Leica Biosystems, Rhenium, Modiin, Israel) was performed with cleaved Caspase 3 (PA5-114687). Tissue viability and assessment of damage to cancer cells were performed according to pathological criteria. The final response score comprising the tumor cell death grading (80%) and the level of damage in live tumor cells (20%) was computed. A scale of 0–100 was created with a score of 0 representing completely viable cancer cells and a score of 100 representing no viable cancer cells.

### 2.15. Statistical Analysis

Data were analyzed and graphed with GraphPad Prism (V.6, Graphpad, San Diego, CA, USA) and expressed as mean ± standard error of the mean (SEM) or standard deviation (STDEV). A *p*-value ≤ 0.05 was considered statistically significant.

## 3. Results

### 3.1. Characterization and In Vitro Activity of ErbB2CAR-T Cells

ErbB2CAR was designed as a second-generation CAR composed of an external human ErbB2-specific scFv and internal T cell co-stimulation and activation domains derived from human CD28 and FcγR, respectively. The ErbB2CAR gene sequence was followed by the combined sequences of an internal ribosome entry site and green fluorescence protein (GFP) in the expression vector (Figure 1A). The ErbB2CAR construct was cloned into the pBullet-based retroviral vector and stably expressed on the T cells isolated from healthy human donors. The expression of CAR was confirmed by flow cytometry based on the co-expressed GFP fluorescence (Figure 1B). The correlation between the GFP and ErbB2CAR expression was previously established by our group [21]. ErbB2CAR was efficiently expressed in T cells with a 50% ± 4.2% transduction on average (Figure 1C). The infected cells were 98.27% ± 0.37% CD3^+^ and comprised 37.95% ± 6.9% CD4^+^ and 51.15% ± 6.4% CD8^+^ cells on average (Figure 1D). The anti-tumor activity of the transduced T cells was evaluated by measuring their direct killing of the human OC cell lines NAR, SKOV3, and OVCAR8. A flow cytometry analysis showed that all three cell lines expressed human ErbB2 to varying extents, with SKOV3 having a median fluorescence intensity (MFI) of one order of magnitude higher than NAR and OVCAR8 (1498.93 vs. 114.44 and 147.22, respectively) (Figure 1E). The ErbB2CAR-transduced lymphocytes (effector cells) were incubated for 16 h with the cancer cell lines (target cells) at different E:T ratios (1:1, 3:1, 10:1, and 30:1), and the killing of target cells was calculated. As shown in Figure 1F, the killing efficiency of the OVCAR8 cells by the ErbB2CAR-T cells was above 60% in all of the tested E:T ratios. However, there was a relatively high non-selective activity of non-infected T cells toward target cells, reaching 60% at the higher E:T ratios. The killing of NAR cells correlated with the E:T ratio, reaching 70% killing at an E:T of 30:1. The killing of SKOV3 cells reached a plateau of approximately 80% at an E:T of 10:1 compared with 10% by the non-infected cells (*p* = 0.0064). The activity of the ErbB2CAR-T cells was further evaluated by measuring the amount of IFNγ secretion, which reflects their ability to undergo specific stimulation when co-cultured with target cells. We found that stimulations of ErbB2CAR-T cells by the target cells were specific but varying: the T cells that were stimulated by SKOV3 secreted 34,440 ± 10,790 pg/mL IFNγ while the T cells stimulated by OVCAR8 and NAR secreted 946 ± 593.26 and 483 ± 265.65 pg/mL IFNγ, respectively. In addition, IFNγ secretion correlated with ErbB2 expression (Figure 1E,G). These results demonstrate the potential anti-tumor activity of ErbB2-directed CAR-T cells towards human OC cells.

### 3.2. Antitumor Effect of Local and Systemic ErbB2CAR Treatment in an OC Mouse Model

To test the efficacy of the ErbB2CAR in vivo, we established an ovarian cancer mouse model, using NSG mice engrafted with human OC cell lines. Based upon the preliminary results that showed the greater aggressiveness of the NAR-derived tumors in mice (compared to OVCAR8 and SKOV3), we used NAR cells in our further in vivo analysis (Appendix A). To enable the spatiotemporal monitoring of the tumor development during the CAR-T cell treatment, we manipulated the tumor cells to express luciferase (NAR-LUC). We compared the systemic to the local CAR-T cell treatments by treating the tumor-bearing mice with ErbB2CAR-T cells administered either intraperitoneally (2.5 × 10^6^ or 10 × 10^6^ cells) or intravenously (10 × 10^6^ cells) 18–23 days following tumor inoculation (Figure 2A). To prevent a partial and variable effect of chemotherapy on the T cells’ viability, the mice were not pretreated with chemotherapy (which is injected IP).

Tumor growth was assessed by bioluminescent imaging (BLI) every 7–10 days. The control groups included untreated or non-infected T cell-treated mice. Figure 2A shows a BLI of a representative experiment. As shown in Figure 2B, all the treatments inhibited tumor progression. The most effective treatment was a high-dose IP administration of ErbB2CAR-T cells (10 × 10^6^ cells), which significantly reduced the tumor burden compared to an IV administration of ErbB2CAR-T cells (10 × 10^6^ cells) and to untreated mice. For example, the radiance on day 52 was 3.97 × 10^6^ ± 1.06 × 10^6^ vs. 1.45 × 10^7^ ± 3.55 × 10^6^ [p/s/cm^2^/sr]. In addition, a low-dose IP administration (2.5 × 10^6^ cells) seems to be more effective than a high-dose IV administration (10 × 10^6^ cells). The treatment efficacy was also assessed by following the survival rates of the treated mice for up to 92 days. Figure 2C shows a Kaplan–Meier survival curve of the three independent in vivo experiments. The averaged median survivals for the untreated mice or mice treated with non-infected T cells were 64 and 63 days, respectively. The median survival was prolonged with the use of a high-dose IP administration of ErbB2CAR-T cells, compared to a high-dose IV administration of ErbB2CAR-T cells (92 vs. 67.5 days, respectively). At 45 days post-NAR-LUC inoculation, the selected mice were euthanized, their ovaries were harvested, and the percentage of ErbB2-positive tumor cells was analyzed by flow cytometry. It emerged that ErbB2 expression decreased significantly in the mice that had been treated with ErbB2CAR administered intraperitoneally, at either a low or high dose, compared to the mice treated with UNT T cells (1.65% and 2.13% vs. 5.64%). In contrast, ErbB2 expression showed no significant decrease in the mice treated with ErbB2CAR administered intravenously compared to the UNT-treated mice (3.07% vs. 5.64%) (Figure 2D). In accordance with the imaging and survival results, we found that ErbB2CAR IP treatments at both low and high doses resulted in a reduction in ErbB2-positive tumor cells in the ovary, and that the outcome was related with low numbers of ErbB2-expressing cells. These results demonstrate that the local administration of CAR-T cells is superior to systemic treatment in terms of tumor elimination and survival in mice, and that the efficacy of ErbB2CAR treatment is dose-dependent.

### 3.3. In Vivo Persistence and Safety Analysis of ErbB2CAR-T Cells

We next evaluated the potential toxicity of the ErbB2CAR-T cells given the possibility that toxicity may be caused by their accumulation in different body organs and by their nonspecific activity toward healthy cells that do not express ErbB2 or that express the antigen at low levels. First, we analyzed the distribution of injected ErbB2CAR-T cells in the spleen, ovary, and blood of the treated mice. At 45 days post-NAR-LUC inoculation, the mice were euthanized, a single-cell suspension was produced from each organ, and the percentage of CAR-T cells (hCD3^+^GFP^+^) was assessed by flow cytometry analysis. As shown in Figure 3A, CAR-T cells were undetectable in the untreated mice as well as in the mice injected with non-infected T cells in all three organs. In contrast, significantly higher levels of CAR-T cells were found in all the organs in the intravenously treated mice compared to both the intraperitoneally treated mice and UNT-treated mice.

We further compared the CAR-T cells’ states before and after injection. The CAR-T cells were stained for CCR7 and CD45RA to analyze the effector to memory cell differentiation. The mice were bled at 45 days after injection and the cells were stained for CCR7 and CD45RA. As shown in Figure 3B, most of the CAR-T cells were naïve before injection and the majority of them were effector memory types after 45 days post treatment; however, no significant changes were detected between the in vivo groups when comparing the different T cell subtypes. Interestingly, a splenic RT-PCR revealed that the exhaustion markers PD-1 and TIM-3 were elevated in the mice treated with ErbB2CAR 10 × 10^6^ cells intraperitoneally compared to the mice treated with ErbB2CAR 10 × 10^6^ cells IV (Figure 3C).

In order to evaluate the potential toxicity of the ErbB2CAR-T cells against normal tissues, we analyzed primary human cell lines derived from healthy essential tissues or tissues known to express ErbB2 according to the Human Protein Atlas site. These were lung epithelial, pancreatic, and cardiac microvascular endothelial cells. The target cells were co-cultured with ErbB2CAR, and their IFNγ secretion levels was measured. As expected, the ErbB2CAR-T cells stimulated by the OC cell lines secreted 28,440 pg/mL and 589 pg/mL of IFNγ upon stimulation by SKOV3 or OVCAR8, respectively. As shown in Figure 3D, ErbB2CAR had residual activity toward most of the tested cells (LEpC-109 pg/mL), but was only moderately stimulated by the pancreatic cells with 242.6 pg/mL of IFNγ secretion. Our in vivo results demonstrate that local CAR-T cell treatment via IP injection limited their distribution to the peripheral blood and their population in the body organs. In addition, ErbB2CAR-expressing T cells demonstrate only minor activity against healthy tissues.

### 3.4. ErbB2CAR Promotes Killing of Primary Ovarian Tumors

Finally, we evaluated the anti-tumor potential of the ErbB2CAR-T cells against primary ovarian tumors. We obtained 29 samples of fresh HGSOCs from women undergoing tumor-debulking surgery. The clinical characteristics of these patients are presented in Table 2. A total of 5 of these 29 patients (number 5, 8, 14, 18, and 25) received neoadjuvant chemotherapy (3 cycles of Carboplatin and Taxol) prior to surgery. A single-cell suspension was produced from each sample and was analyzed for the expression of ErbB2. Interestingly, we found that ErbB2 expression was lower in the tumors from patients pretreated with platinum-based chemotherapy up to two weeks prior to surgery compared to the tumors from women who were chemo-naive (Figure 4A,B). Thirteen of these tissues were then further analyzed by cResponse, and seven were further analyzed as follows. The cResponse^TM^ platform is a novel tool for the prediction of a tumor’s response to various drugs or drug combinations and has been demonstrated to correlate with positive clinical outcomes [22]. The tissues were sliced and incubated with UNT or ErbB2CAR for 72 h. Each analyzed tissue was given a score that indicated the state of the tumor’s histological sections after treatment (see M&M). The supernatant from each section was collected and analyzed for IFNγ secretion. The results imply that the expression levels, the score, and the extent of the IFNγ secretion are interrelated (Figure 4C). The staining of the histological sections for cleaved caspase 3 propagates an apoptotic signal, and the later co-culture with ErbB2CAR-T cells further supported these results (Figure 4D,E). Taken together, these results suggest that ErbB2CAR-T cells have a clinical potential for the treatment of HGSOC. The exact timing of neo-adjuvant treatment proceeding CAR-T treatment should be further evaluated.

## 4. Discussion

In this study, we showed the importance of both the route of administration and the timeline by which EOC patients can be treated with CAR-T cells. We compared local administration (IP) to systemic administration (IV) and assessed the efficacy, as well as the safety, of each method. In a mouse model, we achieved disease remission and prolonged survival when treating the mice with IP ErBb2CAR administration compared to IV. More importantly, intraperitoneally injected ErbB2CAR cells circulated to a lesser extent, making it safer for non-cancerous tissues. Furthermore, we showed that patients that underwent neoadjuvant chemotherapy had lower levels of ErbB2 expression resulting in the decreased efficacy of the ErbB2CAR treatment (Figure 4C). Hence, the treatment sequence is of utmost importance.

Due to the patients’ non-specific presentation and the lack of effective diagnostic tools, more than 75% of EOC patients are diagnosed at advanced stages (stage III to IV) [23]. Moreover, despite improvements in medical therapies [24], particularly with the incorporation of drugs targeting homologous recombination deficiency, EOC survival rates remain low and novel therapeutic approaches are urgently needed [25]. In this study, we have reported a novel therapeutic approach to the treatment of HGSOC.

Immunotherapies are increasingly being explored as a potential treatment option for EOC. Immune checkpoint inhibitors (ICIs) are a traditional class of immunotherapies used to target tumors by disrupting the immune checkpoint pathways enabling immune evasion. ICIs, such as those targeting the PD-1/PDL-1 or CTLA-4 interactions, have shown a high degree of clinical efficacy as single agent therapies in non-small cell lung cancer and melanoma, and the list of cancer subtypes under investigation is rapidly expanding [26]. However, EOC is generally unresponsive to ICI therapy [8,9]. Therefore, our study suggests utilizing another type of immunotherapy, which is a cell-based therapy, in particular CAR-T therapy, which may serve as an alternative adjuvant therapy [8].

ErbB2 is a well-established therapeutic target whose overexpression occurs in various solid tumors, including—but not limited to—breast, gastric, ovarian, colon, bladder, lung, uterine cervix, head and neck, and esophageal cancers [27]. The first ErbB2CAR was constructed and reported by Dr. Eshhar’s group in 1993 [18]. That study demonstrated the feasibility of the construction of the 1st generation ErbB2 CAR containing either the zeta (ζ) chain of the TCR/CD3 complex or the gamma (γ) chain of the immunoglobulin receptor FcεRI. It was later found that the ErbB2CAR-T was more efficient in multiple pre-clinical settings after the addition of co-stimulation molecules. Another analyzed aspect related to ErbB2CAR-T treatment was whether tumors that are resistant to trastuzumab can still be eliminated by ErbB2CAR-T cells redirected by the same antibody domain [28]. The results indicated that CAR-T cells can successfully overcome antibody-resistant tumors by targeting the same epitope, suggesting that CAR-T cells can penetrate the tumor matrix, which acts as a barrier for antibodies.

Currently, there are over 20 ongoing clinical trials on the effect of CAR-T directed at different EOC-associated targets (clinicaltrials.gov). These targets mainly include the proteins mesothelin, Muc16, the alpha folate receptor, and Her2 (ErbB2). Most of the studies concentrate on the systemic delivery of the modified T cells through IV administration [23]. In this study, we aimed to determine whether regional (IP) delivery or systemic (IV) delivery of ErbB2CAR-T cells would be more beneficial for EOC patients. We hypothesized that a regimen that incorporates an IP delivery of treatment would be superior to that of IV administration alone, given that the predominant site of EOC dissemination is within the peritoneal cavity. That rationale and the theoretical benefits for the direct delivery of IP treatment into the peritoneal cavity was already acknowledged decades ago [29]. IP chemotherapy for peritoneal metastases offers clear pharmacokinetic and pharmakodinamic advantages, mainly by enabling high loco-regional drug concentrations s concomitantly with low systemic drug concentrations. Therefore, it is reasonable to hypothesize that the IP delivery of modified CAR-T cell therapy may also lead to improved efficacy. This concept is currently being evaluated in a phase I dose escalation trial for patients with recurrent MUC-16(ecto^+^) OC testing the safety of IV and IP administrations and the preliminary efficacy of autologous IL-12 secreting MUC-16 (ecto) CAR-T cells containing a safety-elimination gene [30].

In the study by Chekmasova et al. [31], SCID-Beige mice, injected IP with human OV-CAR3 tumor cells and then treated either IV or IP with MUC16CAR T cells, showed significantly improved survival compared to the control mice, which were either untreated or treated with CAR-T cells targeting the irrelevant antigen CD19. The intravenously and intraperitonially treated mice exhibited statistically equivalent antitumor efficacies using the second-generation CAR (MUC16-28z). However, the safety evaluation remains in question with regard to IV versus IP administrations.

Management of all tumor sites, is more challenging when surgery fails and the cancer has not been completely removed. Combining CAR-T IP delivery with the surgical removal of the tumor may target and eliminate those hidden remaining cells that ultimately lead to disease recurrence. This method can be beneficial to patients whose tumor is limited to the abdomen while both IV and IP administrations will likely achieve optimal responses in patients for whom metastatic disease is present outside of the peritoneum.

ErbB2 CAR-T cells pose a risk of lethal toxicity, including cytokine release syndrome resulting from the “on-target, off-tumor” recognition of ErbB2. Morgan et al. [32] presented a case report of ErbB2 CAR-T administered to a patient with colon cancer that metastasized to the lungs and liver. The patient experienced respiratory distress and displayed a dramatic pulmonary infiltrate on a chest X-ray within 15 min after cell infusion. The patient was intubated and passed away 5 days after treatment despite intensive medical intervention. Later on, investigators at Baylor College of Medicine conducted various trials to determine the safety and antitumor efficacy of HER2-CAR CMV-CTLs and showed a successful generation and evaluation in 16 HER2^+^ glioblastoma patients (NCT01109095). The latter clinical study and more recent ones were conducted with a CD28 co-stimulation molecule (Table 1) [27]. Furthermore, in order to determine the toxicity in the pediatric population, a loco-regional phase 1 clinical trial at Seattle Children’s Hospital evaluated the repetitive dosing of HER2-specific CAR-T cells in patients with recurrent/refractory central nervous system (CNS) tumors, including diffuse midline glioma [33]. The initial three patients experienced no dose-limiting toxicity and exhibited clinical as well as correlative laboratory evidence of local CNS immune activation, including high concentrations of CXCL10 and CCL2 in the cerebrospinal fluid.

There have been several attempts to build safer ErbB2 CAR-T cells. One example is either by the means of an affinity-tuned scFvs [34] or a “dual targeting CARs” approach [35,36]. Another way to ensure the safe use of CAR-T cells is to incorporate an inducible caspase 9 suicide gene within the cells [37]. A further novel design involves engineering co-stimulated HER2 CAR-T cells to co-express a PD-1- or CTLA4-based inhibitory CAR that recognizes other members of the HER family. These strategies may help overcome the potential toxicity of ErbB2 CAR. Still, CAR-T cells directed to ErbB2 are monitored carefully and dose escalation is evaluated [38].

Based upon our current study’s findings, we propose that the local administration (IP) of ErbB2CAR is safer than systemic administration (IV) with respect to on-target, off-tumor toxicity. We observed a significantly higher presence of CAR-T cells in the blood, spleen, and ovary of EOC mice 45 days post-systemic administration compared to a local administration (Figure 3A). These results further support the hypothesis that on-target, off tumor toxicity with ErbB2CAR is more likely to occur with systemic injections than with local injections. Importantly, we show that the IP route is not only safer but also more efficient than the IV route (median survival of 92 vs. 67.5 days).

Efficient CAR-T function depends on the expression level of the antigen presented on the tumor cells. We, in collaboration with cResponse^TM^, have shown that prior chemotherapy administration in OC patients leads to the down-regulation of ErbB2 expression in the tumor (Figure 4A). Personalized treatments have now become well accepted. Individual tumors can be analyzed by deep sequencing and proteomics to define better treatment options that combine specific monoclonal antibodies or other targeted therapeutics. This analysis can help navigate patient-specific treatments; however, the prediction of improved responses to combined treatments is still lacking. Utilizing the cResponse^TM^ platform, we showed that tissues from HGSOC patients undergoing chemotherapy express lower levels of HER2 resulting in a reduced response to the ErbB2CAR. These results emphasize the importance of analyzing the tumor before choosing the treatment protocol and the cruciality of the sequence in which the treatments are given to the patient. Thus, the concept of an OC IP administration as a local and not a systemic treatment is not only safer but may also increase efficiency in the setting of low ErbB2 expression.

## 5. Conclusions

We have shown that ErbB2CAR-T may serve as a potential treatment modality for HGSOC with diminished toxicity and higher efficacy when administered locally (IP). We further showed that an administration of IP low-dose ErbB2CAR-T is better than an administration of a high dose intravenously. Our current results highlight the value of accurate prediction responses for combinational therapy for patients with HGSOC and propose a novel comprehensive approach towards EOC treatment combining surgical treatment with ex-vivo analysis of the tumor followed by CAR-T treatment.

## Figures and Tables

**Figure 1 biomedicines-10-02216-f001:**
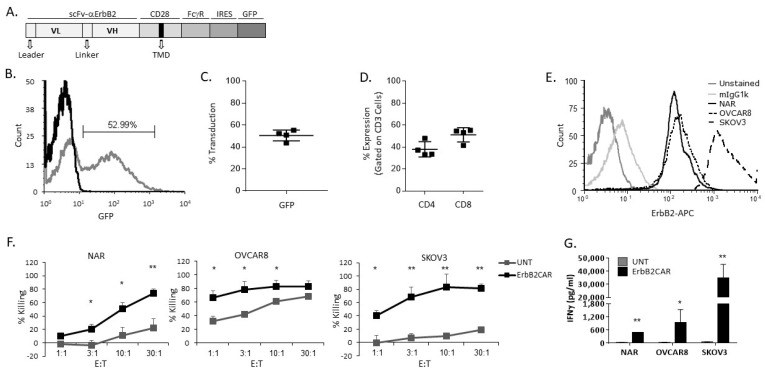
**Characterization and in vitro activity of ErbB2CAR-T cells**: (**A**) The ErbB2CAR construct is comprises the ScFv specifically directed against ErbB2 followed by the co-stimulation domain of CD28 and the activation domain from FcγR. (**B**,**C**): Retroviral supernatant collected from ErbB2-expressing packaging cells (PG13) was used to infect PBLs from a healthy donor. Infected and non-infected cell cultures were grown for an additional 3 days following transduction. On the day they were harvested, the cells were analyzed by flow cytometry and the CAR expression level was calculated. (**B**) A representative flow cytometry analysis of lymphocytes transduced with ErbB2CAR-GFP. (**C**) Average transduction expression of the ErbB2CAR-GFP. The dots indicate four independent experiments, and the lines show mean ± STDEV. (**D**) Expression of CD4/CD8 gated from CD3 positive cells out of ErbB2CAR-GFP transduced T cells. Transductions of four donors are shown; a line indicates the average. (**E**) Expression of ErbB2 antigen on the human OC cell lines NAR, OVCAR8, and SKOV3. Cells were stained with anti-human ErbB2-APC or mouse IgG1k-APC isotype control and analyzed by flow cytometry. (**F**) Killing assay using effector (E) and target cells (T). Cells were incubated for 16 h with ErbB2CAR or UNT T cells (as control) at different E:T ratios as indicated. The percentage of killing was measured by methylene blue colorimetric assay. An average of five ErbB2CAR versus UNT experiments are presented with ± SEM. * *p* < 0.05; ** *p* < 0.01. (**G**) T cells transduced with ErbB2CAR (black bars) or UNT cells (gray bars) were co-cultured with the indicated ovarian cell lines at an E:T ratio of 2:1. After 16 h, IFNγ levels in culture supernatants were measured by ELISA. The results shown are the average of three experiments ±STDEV. ErbB2CAR versus UNT * *p* < 0.05; ** *p* < 0.01. F/G statistical analysis was performed using Student’s *t*-test between ErbB2CAR versus UNT treatment.

**Figure 2 biomedicines-10-02216-f002:**
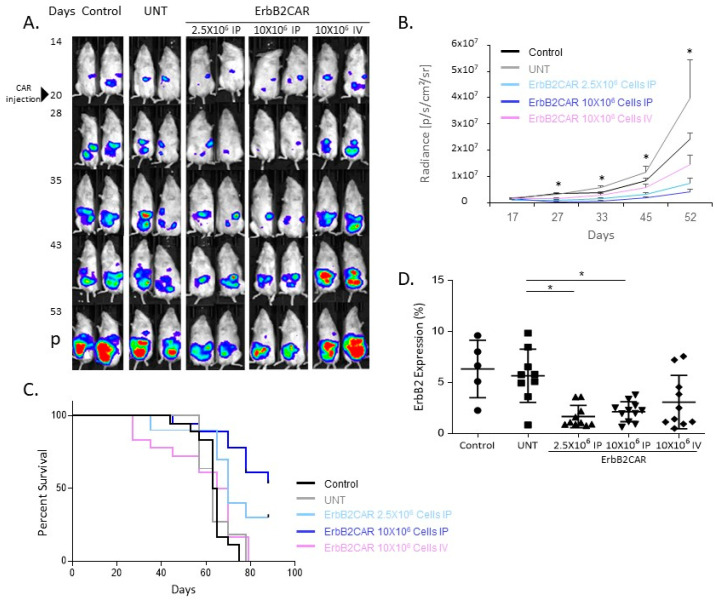
**Antitumor effect of ErbB2CAR in an ovarian mouse model.** Mice were intraperitoneally engrafted with 2 × 10^6^ luciferase-expressing NAR tumor cells (NARLUC). After 20 days, the mice were treated with ErbB2CAR (2.5 × 10^6^ IP or 10 × 10^6^ IP or 10 × 10^6^ IV), with non-infected T cells (UNT), or left untreated (Control). (**A**) Tumor burden assessment of NARLUC in treated mice as imaged by IVIS at the indicated days. The arrowhead indicates CART injection at day 20. (**B**) Summary of averaged radians (p/s/cm^2^/sr) of three in vivo experiments. Each group includes 10–18 mice ± SEM. Data were analyzed by 2-way ANOVA test of ErbB2CAR (2.5 × 10^6^ IP or 10 × 10^6^ IP) versus UNT * *p* < 0.01, and 10 × 10^6^ IP versus 10 × 10^6^ IV * *p* < 0.05 ErbB2CAR 10 × 10^6^ IV versus UNT. (**C**) Kaplan–Meier survival curves of mice in different treatment groups. Each group included 10–18 mice ± SEM. * *p* < 0.0001 ErbB2CAR 10 × 10^6^ IP versus 10 × 10^6^ IV and ErbB2CAR 10 × 10^6^ IP versus UNT. * *p* < 0.05 ErbB2CAR 10 × 10^6^ IP versus UNT. ErbB2CAR 10 × 10^6^ IV versus UNT was not significant. (**D**) The mice were euthanized 45 days post-NAR-LUC injection. Cells from ovaries were isolated, stained with anti-human ErbB2-APC, and analyzed by flow cytometry. Each group includes an average of 5–11 mice ± SEM. Data were analyzed by 1-way ANOVA test * *p* < 0.05.

**Figure 3 biomedicines-10-02216-f003:**
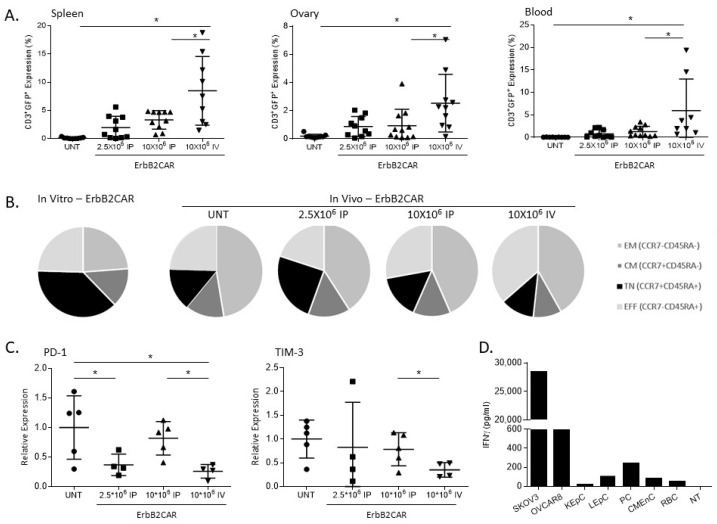
**In vivo persistence and safety analysis of ErbB2CAR-T cells.** (**A**) The mice were injected with 2 × 10^6^ cells IP NAR-LUC cells. After 20 days, they were treated with ErbB2CAR (2.5 × 10^6^ IP or 10 × 10^6^ IP or 10 × 10^6^ IV cells), UNT, or were left untreated. They were euthanized after 45 days. Cells from the spleen, ovary, and blood were collected and stained with anti-human CD3 and analyzed by flow cytometry. Each group included an average of 8–11 mice ± SEM. Data were analyzed by one-way ANOVA test * *p* < 0.05. (**B**) The leftmost pie diagram depicts the percentage of CCR7 and/or CD45RA expression out of CD3/ErbB2CAR-GFP-positive T cells (average of four independent transductions). The mice were then euthanized 45 days post-NAR-LUC injection. Splenic single-cell suspension was prepared and stained with anti-human CD3/CCR7/CD45RA and analyzed by flow cytometry. The three right pies depict the percentage of CCR7 and/or CD45RA expression out of CD3-positive cells. EM—effector memory cells, CM—central memory, TN—naïve T cells, and EFF—effector T cells. Each group included an average of 5–11 mice. (**C**) The mice were euthanized 45 days post-NAR-LUC inoculation, and RNA levels from the splenic cells were measured for PD-1 and TIM-3. The relative expressions were normalized to the housekeeping gene GAPDH, and UNT-treated mice were considered as 1. Each group included an average of 4–5 mice ± SEM. Data were analyzed by 1-way ANOVA test * *p* < 0.05. (**D**) Primary cells from different tissues of healthy human donors were co-cultured with lymphocyte T cells transduced for 24 h with ErbB2CAR at an E:T ratio of 2:1. Supernatants were collected and secreted IFNγ was measured by ELISA. The tested target cells included: SKOV3 and OVCAR8 cells (positive control), KEpC—Kidney Epithelial Cells, LEpC—Lung Epithelial Cells, PC—Pancreatic Cells, CMEnC—Cardiac Microvascular Endothelial Cells, RBC—Red Blood Cells, and NT—No Target.

**Figure 4 biomedicines-10-02216-f004:**
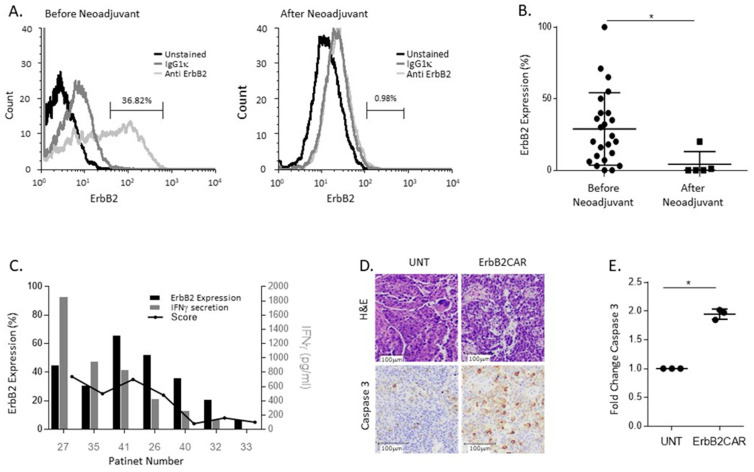
**ErbB2CAR specifically promotes killing of primary ovarian tissues.** Tissues were collected from patients with HGSOC before and after neoadjuvant treatment and signal cell suspension was stained with anti-human ErbB2-APC. mIgG1-APC was used for isotype control. The cells were analyzed by flow cytometry. (**A**) A representative flow cytometric analysis of samples from patients before neoadjuvant treatment (**left**) and after neoadjuvant treatment (**right**). (**B**) Summary of ErbB2 expression in 24 patients before neoadjuvant and 5 patients after neoadjuvant as analyzed by flow cytometry. Data were analyzed by Student’s *t*-test before versus after neoadjuvant treatment * *p* < 0.05. (**C**) Seven tissues were randomly selected for further analysis by cResponse^TM^. Single cells were stained for anti-human ErbB2 expression determined by flow cytometry (black bars). Histological sections were co-cultured for 72 h with ErbB2CAR or UNT cells. Supernatants were collected and analyzed for IFNγ secretion as determined by ELISA (gray bars), and H and E staining determined tissue viability after treatment. The score was calculated as described in Materials and methods. The viability of the cancer cells was calculated as the delta score of ErbB2CAR treatment and of UNT treatment (black line). (**D**) Representative staining with H&E (top panel) and cleaved Caspase 3 (bottom panel) of HGSOC tissue after co-culture with ErbB2CAR or UNT cells for 72 h. (**E**) Fold change in cleaved Caspase 3-positive cells as calculated from the average of four areas in each tissue. UNT treatment was considered as 1. The displayed results are the average of three tissues ± STDEV. Data were analyzed by Student’s *t*-test ErbB2CAR versus UNT treatment * *p* < 0.01.

**Table 1 biomedicines-10-02216-t001:** Primer sequences used for amplification.

Gene	Forward Primer 5′ to 3′	Reverse Primer 5′ to 3′
PD-1	CCAGGATGGTTCTTAGACTCCC	TTTAGCACGAAGCTCTCCGAT
TIM-3	AGACAGTGGGATCTACTGCTG	CCTGGTGGTAAGCATCCTTGG
GAPDH	AGGGCCCTGACAACTCTTTT	TTACTCCTTGGAGGCCATGT

**Table 2 biomedicines-10-02216-t002:** Patients’ Characterization: Table contains patient numbers, age, grade, and tumor histology: HGSOC/high grade serous carcinoma, surgery—primary debulking, interval debulking ± HIPEC, or secondary debulking; chemotherapy—Carboplatin + Taxol ± Avastin.

Patient Number	Age	Grade	Tumor Histology	Surgery	Chemotherapy
1	67	3	HGSOC	Primary Debulking	Carbo Taxol
2	65	3	HGSOC	Interval Debulking	Carbo Taxol
3	59	3	HGSOC	Interval Debulking	Carbo Taxol
4	28	3	HGSOC	Primary Debulking	Carbo Taxol + Olaparib
5	71	3	HGSOC	Primary Debulking	Carbo Taxol + Avastin
6	48	3	HGSOC	Primary Debulking	Carbo Taxol + Avastin
7	61	3	HGSOC	Interval Debulking	Carbo Taxol
8	68	3	HGSOC	Primary Debulking	Carbo Taxol + Avastin
9	71	3	HGSOC	Interval Debulking + Hipec	Carbo Taxol
10	47	3	HGSOC	Primary Debulking	Carbo Taxol
11	63	3	HGSOC	Interval Debulking + Hipec	Carbo Taxol
12	47	3	HGSOC	Interval Debulking	Carbo Taxol
13	56	3	HGSOC	Interval Debulking	Carbo Taxol + Avastin
14	69	3	HGSOC	Primary Debulking	Carbo Taxol
15	64	3	HGSOC	Interval Debulking	Carbo Taxol + Avastin
16	60	3	HGSOC	Primary Debulking	Carbo Taxol
17	66	3	HGSOC	Interval Debulking	Carbo Taxol
18	67	3	HGSOC	Interval Debulking	Carbo Taxol
19	83	3	HGSOC	Primary Debulking	No Chemotherapy
20	73	3	HGSOC	Primary Debulking	Carbo Taxol
21	67	3	HGSOC	Interval Debulking	Carbo Taxol + Avastin
22	56	3	HGSOC	Interval Debulking	Carbo Taxol + Avastin
23	53	3	HGSOC	Interval Debulking	Carbo Taxol
24	46	3	HGSOC	Interval Debulking	Carbo Taxol + Avastin
25	62	3	HGSOC	Interval Debulking	Carbo Taxol + Avastin
26	59	3	HGSOC	Secondary Debulking	
27	76	3	HGSOC	Interval Debulking	Carbo Taxol
28	70	3	HGSOC	Interval Debulking	Carbo Taxol + Avastin
29	75	3	HGSOC	Interval Debulking	Carbo Taxol + Avastin

## Data Availability

Not applicable.

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
