# Peer review of "Comparing Intraperitoneal and Intravenous Personalized ErbB2CAR-T for the Treatment of Epithelial Ovarian Cancer"

_biomedicines, 2022, doi:10.3390/biomedicines10092216_

Round 1

Reviewer 1 Report

Authors have presented a well-structured research article emphasizing “Intraperitoneal Personalized ErbB2CAR-T Treatment for Epithelial Ovarian Cancer”. The article is original, well structured; easy to read with main emphasis on intraperitoneal (IP) versus intravenous (IV) CAR-T cell therapy in the treatment of high-grade ovarian cancer. In my opinion, the manuscript can be published in this journal, after the authors have addressed the following issues:

·       Authors are suggested to improve the title of the manuscript as it seems to be a comparative study between IP vs. IV model of CAR-T cell therapy, so correlate with this.

·       Please explain the reason of selecting only two markers viz. PD-1 and TIM-3.

·       What type of statistical analysis was performed for in vivo and clinical studies. Mention clearly in methodology section.

·       Authors have selected 29 clinical samples for their study. Please explain the reason.

·       In discussion section, line number 445-448 related to ICIs, authors can add some relevant information from recent review article: https://doi.org/10.3390/ph15030335

·       Authors should improve the introduction section by adding some content highlighting the role of ErbB2 for the better understanding of readers and to clear the concept formulation.

·       The study designed is quite vast and various parameters are used to correlate the findings, if possible, add a summary diagram or graphical abstract.

·       Grammatical and formatting issues are there in manuscript at several places for example, no subsections numbering, line number 50, 57, 142……, authors are suggested to carefully proofread the manuscript to eliminate such type of errors.

·       Add a separate section for conclusion that explain the main findings and future prospects of current study in well-defined manner.

After incorporating these suggestion, the manuscript can be considered for publication.

Author Response

We highly appreciate the reviewer comments, we took it into our attention and made some crucial changes. 

1. Authors are suggested to improve the title of the manuscript as it seems to be a comparative study between IP vs. IV model of CAR-T cell therapy, so correlate with this.

We thank the reviewer, we have modified the title to reflect the main narrative of our study. The title was changed to : Comparing Intraperitoneal and Intravenous Personalized ErbB2CAR-T for the Treatment of Epithelial Ovarian Cancer

2.       Please explain the reason of selecting only two markers viz. PD-1 and TIM-

The reviewer raised a good point, we have tested other exhaustion markers as CTLA-4 and LAG3. However, these results were not significant thus were not further analyzed and presented.

3.  What type of statistical analysis was performed for in vivo and clinical studies. Mention clearly in methodology section.

Thank you for this crucial point, we have modified the legends, and added the exact statistic test that has been done.

4.   Authors have selected 29 clinical samples for their study. Please explain the reason.

Our experimental system requires fresh tissues 29 is the total number of suitable fresh human samples that could be collected in our tertiary medical center over 2 year time period.

5.   In discussion section, line number 445-448 related to ICIs, authors can add some relevant information from recent review article: https://doi.org/10.3390/ph15030335

We thank the reviewer for bringing this recent review to our attention, we carefully read it and incorporated it in to our revised discussion.  

6.   Authors should improve the introduction section by adding some content highlighting the role of ErbB2 for the better understanding of readers and to clear the concept formulation.

We agree with the reviewer, we added a paragraph to the introduction explaining the roll of ErbB2 in ovarian cancer.

7. The study designed is quite vast and various parameters are used to correlate the findings, if possible, add a summary diagram or graphical abstract.

We agree with the reviewer, thus we have added clarifications in the text. However, we didn’t  add an graphical abstract as this was not feasible for our study

8.  Grammatical and formatting issues are there in manuscript at several places for example, no subsections numbering, line number 50, 57, 142……, authors are suggested to carefully proofread the manuscript to eliminate such type of errors.

We agree with the reviewer, thus we clarified and explained better throughout the article.

9.   Add a separate section for conclusion that explain the main findings and future prospects of current study in well-defined manner.

We agree with the reviewer, thus we added after the discussion.

After incorporating these suggestion, we hope that you will consider our  article  for publication

Sincerely, 

Anat Globerson Levin

Reviewer 2 Report

The major useful finding of the study is that IP route for CAR-T may be adavantageous in some circumstance as compared to IV.

Some issues need improvement:

cResponce is not explaind sufficiently. I cannot find a link to that product.

Is there (if not -why?) GvH caused by human T cells in mice?

The limitations of the study must be acknowledged. This is a allogeneic CAR-T in a nude mice background ... Very far from any clinical scenario.

The controls throughout the paper a not great. The appropriate control for CAR transfected T cell would be vector transfected or even better CD28-transfected T cell. There is HLA disparity so this must be well proved the killing is Erbb2 mediated.Can the authors defend the choice and the reliability of the controls

Do the authors have any specific marker for their CAR-T? Do all the persisting T cell express CAR?

Please explain how paragraph 3.4 justify this claim 'ErbB2CAR specifically promotes killing of primary ovarian tissues'?

Author Response

We highly appreciate the reviewers comments. We have applied all comments in the revised article as follows: 

  1. cResponce is not explaind sufficiently. I cannot find a link to that product.

We agree with the reviewer and added a short explanation in the method section. A more detailed explaining on cResponce platform is referenced at the appropriate paragraph in the results

  1. Is there (if not -why?) GvH caused by human T cells in mice?

Reviewer is correct. As one of the most established laboratories for CAR T cell therapy, our observation is  that models using NSG mice develop GvH 40-50 days  following human T cells administration. This is also implied in our survival analysis figure, however as can be concluded, it  is not lethal.

  1. The limitations of the study must be acknowledged. This is a allogeneic CAR-T in a nude mice background ... Very far from any clinical scenario.

The reviewer is correct. To overcome this limitation we have combined several approaches  to better predict the clinical application. The NSG model – a widely accepted animal model  for CAR T cell therapy, was combined with cResponse system to better evaluate the effects of the treatment in human environment.

  1. The controls throughout the paper a not great. The appropriate control for CAR transfected T cell would be vector transfected or even better CD28-transfected T cell. There is HLA disparity so this must be well proved the killing is Erbb2 mediated. Can the authors defend the choice and the reliability of the controls

To our best understanding , the reviewer recommends activated T cells as control. As mentioned in the M&M section, the process of transduction, is initiated with T cells activation using anti-CD3 and anti CD28. Our control even though not transduced are also activated T cells using the mentioned antibodies. This is an accepted and widely used  control in previously CAR T publications. 

  1. Do the authors have any specific marker for their CAR-T? Do all the persisting T cell express CAR?

  1. Please explain how paragraph 3.4 justify this claim 'ErbB2CAR specifically promotes killing of primary ovarian tissues'?

Reviewer is correct, we have revised the title of the paragraph to better reflect the results of our study in human ovarian cancer samples. The title was changed to:ErbB2CAR promotes killing of primary ovarian tumors

We hope that after incorporating these suggestion, the reviewer will consider our article for publication.

Sincerely,

Anat Globerson Levin

Round 2

Reviewer 1 Report

Authors have revised the manuscript in very well manner, now the manuscript can be considered for publication in its present form.

Reviewer 2 Report

The manuscript is improved now.